# Involvement of Intestinal Enteroendocrine Cells in Neurological and Psychiatric Disorders

**DOI:** 10.3390/biomedicines10102577

**Published:** 2022-10-14

**Authors:** Liangen Yu, Yihang Li

**Affiliations:** 1Department of Animal and Food Sciences, University of Delaware, Newark, DE 19716, USA; 2Department of Biological Sciences, University of Delaware, Newark, DE 19716, USA

**Keywords:** enteroendocrine cells, enterochromaffin, GLP1, GLP2, serotonin, Parkinson’s disease, schizophrenia, visceral pain, depression, gut–brain axis

## Abstract

Neurological and psychiatric patients have increased dramatically in number in the past few decades. However, effective treatments for these diseases and disorders are limited due to heterogeneous and unclear pathogenic mechanisms. Therefore, further exploration of the biological aspects of the disease, and the identification of novel targets to develop alternative treatment strategies, is urgently required. Systems-level investigations have indicated the potential involvement of the brain–gut axis and intestinal microbiota in the pathogenesis and regulation of neurological and psychiatric disorders. While intestinal microbiota is crucial for maintaining host physiology, some important sensory and regulatory cells in the host should not be overlooked. Intestinal epithelial enteroendocrine cells (EECs) residing in the epithelium throughout intestine are the key regulators orchestrating the communication along the brain-gut-microbiota axis. On one hand, EECs sense changes in luminal microorganisms via microbial metabolites; on the other hand, they communicate with host body systems via neuroendocrine molecules. Therefore, EECs are believed to play important roles in neurological and psychiatric disorders. This review highlights the involvement of EECs and subtype cells, via secretion of endocrine molecules, in the development and regulation of neurological and psychiatric disorders, including Parkinson’s disease (PD), schizophrenia, visceral pain, neuropathic pain, and depression. Moreover, the current paper summarizes the potential mechanism of EECs in contributing to disease pathogenesis. Examination of these mechanisms may inspire and lead to the development of new aspects of treatment strategies for neurological and psychiatric disorders in the future.

## 1. Introduction

The number of patients suffering from neurological and psychiatric disorders has increased dramatically in the past few decades. According to the World Health Organization (WHO) epidemiology statistics, the number of Parkinson’s disease (PD) patients has doubled within the last 25 years [1]. Moreover, recent updates from the WHO indicate that there are nearly a billion people suffering from mental disorders, while approximately 280 million people suffer from depression around the world [2,3]. However, effective treatments for neurological and psychiatric disorders are currently limited due to the heterogeneous disease pathogenesis and targets of treatments. For example, it has been shown that visceral pain or depression patients sometimes express resistance to treatment [4,5,6,7]. Therefore, in order to develop novel therapeutic strategies, the exploration of novel aspects of neurological or psychiatric pathogenic mechanisms is urgently required.

Enteroendocrine cells (EECs) are chemosensory cells residing in the intestinal epithelium, and they function as important sensors monitoring changes in the lumen of the gastrointestinal (GI) tract. The EECs orchestrate not only the communication with luminal microorganisms via microbial metabolites, but also the communication with host body systems via neuroendocrine hormones (Figure 1 and Figure 2). For example, epithelial EECs continuously respond to short-chain fatty acids (SCFAs) generated by luminal microorganisms via free fatty acid receptor 2 and 3 (FFAR2/3). Following this transceptor or receptor activation, EECs secrete pre-made peptide hormones to conduct paracrine and endocrine functions [8,9,10,11]. Further, the neuropod structure of EECs allows direct or indirect signal transductions to enteric glia cells and enteric neurons [12,13]. An increasing body of evidence indicates the unique involvement or pathogenic role of the gut–brain axis and gut microbiota in neurological and psychiatric disorders, in which EECs might participate [14,15,16,17,18,19].

The present paper reviews the current understanding and newly published evidence regarding how EECs and their peptide hormones are involved in neurological and psychiatric disorders. Moreover, the current paper focuses on the mechanism of EECs to summarize the potential pathways for the development of new aspects of treatment strategies in the future. The papers in the present review were generally selected under the scope of EECs and endocrine molecules including serotonin, glucagon-like peptide 1 (GLP1), glucagon-like peptide 2 (GLP2), peptide YY (PYY), as well as their involvement in particular neurological psychiatric disorders, such as PD, schizophrenia, visceral pain, neuropathic pain, and depression.

## 2. Enteroendocrine Cell Functions That Might Be Related to Neurological and Psychiatric Disorders

EECs are located in the epithelium throughout the GI tract. They dynamically produce and store various peptide hormones and bioactive components, depending on the intestinal segments and epithelial homeostasis status. The regulation of EECs’ content profile, as well as their functions in energy metabolism and roles as incretins, has been reviewed elsewhere [20,21]. EECs can be further categorized into multiple subtypes, depending on their endocrine molecules production and secretion. For example, G cells can be identified by the secretion of gastrin; K cells uniquely secrete gastric inhibitory peptides; L cells produce and secrete GLP1, GLP2, PYY, and oxyntomodulin; I cells produce cholecystokinin (CCK); N cells secrete neurotensin; S cells secrete secretin; enterochromaffin cells (ECs) secrete serotonin; and enterochromaffin-like cells secrete histamine [20,21]. Here, we only focus on the subtypes of EECs that are potentially involved in neurological and psychiatric disorders.

L cells mostly secrete GLP1, GLP2, PYY, and oxyntomodulin, although PYY might also be co-expressed with gastrin, which is mostly secreted by G cells [22]. GLP1, GLP2, and PYY, which are secreted by L cells, and serotonin, secreted predominantly by ECs (Figure 2), are discussed in this paper. GLP1 and GLP2 have been shown to correlate with multiple neurological disorders. Serotonin also showed a correlation with depression and visceral pain, although this is still under debate. Besides incretin functions, GLP1 has been shown to exert anti-inflammatory effects in both the GI tract and central nervous system (CNS) [23,24,25]. Moreover, GLP1 possesses neuroprotective effects and triggers neurogenesis [26,27,28,29,30,31]. New evidence suggested that GLP1 and glucagon-like peptide 1 receptor (GLP1R), a receptor of GLP1, have protective effects on hypothalamic inflammation and leptin sensitivity in mice [25,32]. Despite the well-known source and their effects within the CNS, the GLP1 derived from intestinal EECs has also been suggested to play a role in neurological pathology, due to the feature wherein GLP1 is able to pass through the brain–blood barrier [33,34,35]. Similar to GLP1, L-cell-secreted GLP2 also possesses anti-inflammation effects [36,37,38]. In cows, GLP2 administration increased the intestinal villi height, mucosal surface, and proliferating cells, and decreased inflammation [39]. Further, GLP2 has a neuroprotective effect and can trigger neurogenesis in a similar manner as GLP1 [29,37,40,41,42]. Interestingly, the anti-inflammatory effects of other components of EEC content have recently been revealed, including PYY [43,44] and serotonin [45,46], which are likely associated with neuroinflammation. Therefore, accumulating evidence suggests that EECs and ECs could play important pathogenic and regulatory roles in neurological and psychiatric disorders.

## 3. Enteroendocrine Cells in Parkinson’s Disease

PD is a common movement disorder that was originally characterized as a neurodegenerative disorder due to the loss of dopaminergic neurons and accumulated aggregation of α-synuclein fibrils (called Lewy bodies) (reviewed elsewhere previously [47]). However, studies have shown a new pathogenic aspect of PD, which could be linked to intestinal disorders, as well as to changes in intestinal microbiota and metabolites [15,48,49]. For instance, inflammatory bowel disease (IBD) has increased by 22 to 35% regarding the incidence of PD [50]. In addition, Sampson et al. reported that the GI microbiota was required for motor deficits, microglia activation, and α-synuclein pathology (PD symptoms), in a germ-free mice model overexpressing α-synuclein. Further, their results indicated that the microbial metabolites produced in PD patients enhanced the pathophysiology of PD [15]. Although with a negative correlation, others also found an association among the GI microbiota, the total faecal SCFAs, and PD incidence [48]. Researchers hypothesized that the origin of PD might lie in the enteric nervous system (ENS) [51,52]. Accordingly, α-synuclein was detected in GI mucosa in early PD patients [53,54].

Given the important luminal chemo-sensing and neuroendocrine functions of EECs, these recent results point to a hypothesis that EECs contribute to and regulate the pathogenesis of PD. Interestingly, in human intestinal tissue, the α-synuclein that triggers PD was colocalized with EECs, such as L cells and K cells [55,56]. Although the authors have not confirmed the original secretion location of the α-synuclein, the data in these studies strengthen the possibility of EECs’ involvement in PD progression.

Two potential mechanisms of EECs’ contribution in PD pathogenesis have been proposed. On one hand, the EECs are likely to be a source of α-synuclein, which is generated in response to specific microbial activation. Thereafter, the α-synuclein is transported into the brain via nerves, leading to the accumulation of α-synuclein [57] (Figure 3a). In line with this hypothesis, a very recent research work revealed the potential mechanisms. The authors identified an increased population of microorganism *Akkermansia muciniphila* in the guts of PD patients. The metabolites of this microorganism initiated α-synuclein aggregation in EECs, via activation of ryanodine receptor (RyR), calcium ion (Ca^2+^) release, and increased mitochondrial reactive oxygen species (ROS) generation [58] (Figure 3b). Moreover, a newly published paper indicated that another microbial metabolite, sodium butyrate, increased the α-synuclein mRNA expression in EECs through the autophagy-related 5 (Atg5) dependent autophagy pathway [59]. Holmqvist et al. provided evidence that α-synuclein was able to move from the intestine to the brain in rats [60]. Further, the transportation of α-synuclein from EECs to neurons requires GTPase called Ras-related protein Rab-35 (Rab35) and cell-to-cell contact, which is in line with the EECs’ characteristics [61].

On the other hand, the EECs’ secretion could also be suppressed by alterations in luminal SCFA concentrations and profiles. This could be the consequence of changes in specific microbes, which then increase the systemic inflammation, and this eventually enhances the progression of PD [62,63,64]. It was suggested that sodium butyrate increased the pro-inflammatory cytokines and α-synuclein mRNA expression in an EECs cell line and neuroblast cell line treated with EECs conditional medium [59]. Further, the EECs facilitate α-synuclein transport, which could trigger inflammation responses in microglia [65,66]. In contrast, a study in a PD mouse model suggested that the oral administration of butyrate could have protective effects on the neurobehavioral impairment via increased EEC activities, such as increased colonic GLP1 expression and brain GLP1R gene expression [67]. A recent animal study also indicated the neuroprotective effect of GLP1, triggered by chlorogenic acid [31]. These conflicting characteristics of EECs might be due to the variations in EECs’ homeostasis status or the hormone composition of EECs. In other words, the EECs that secrete GLP1 could be beneficial in terms of inflammation reduction, while the EECs that cannot secrete GLP1 but produce α-synuclein could be harmful. However, the detailed mechanism for either hypothesis is still unclear, especially regarding the extent to which EECs contribute to inflammation in PD patients. Future study will be needed to investigate the detailed mechanisms of EECs in PD progression.

## 4. Enteroendocrine Cells in Schizophrenia

Schizophrenia is a complex neurodevelopmental disorder that could be significantly defined by observations of psychosis signs. In most cases, schizophrenia patients present paranoid delusions and auditory hallucinations [68]. Due to the complexity of neurodevelopment and schizophrenia, the mechanism behind schizophrenia remains unknown [16].

Schizophrenia has been suggested to be associated with the impaired function and structure of synapses [69,70]. Moreover, the microbiota is also associated with synaptogenesis and synapse maturation [16,71]. While the dietary manipulation and intestinal SCFAs enhancement in schizophrenia patients have been discussed elsewhere [72], there have been fewer connections identified between schizophrenia and luminal SCFAs and EECs.

A recent study provided new evidence that linked schizophrenia and epithelial EECs [73]. Uellendahl-Werth et al. reported the susceptibility genes shared between EECs and schizophrenia. Their results indicated that protein phosphatase 3 catalytic subunit alpha (PPP3CA) is the shared susceptibility locus for IBD (Crohn’s disease and ulcerative colitis) and schizophrenia. The genes were expressed in restricted tissues, including neurons in the brain, intestinal epithelial EECs, and Paneth cells in the ileum, colon, and rectum [73]. The authors also provided two possible mechanisms by which PPP3CA in EECs contributes to disease pathology. First, EEC modulation altered the neuronal signal transduction in the striatum. Second, the EECs modulated inflammation responses. Several studies discussed the beneficial effects of the GLP1 (EECs product) agonists on metabolic disorders in schizophrenia patients [74], while others also revealed the potential neuroprotective effect of GLP1 agonists [75,76]. In contrast, several controversial data also suggested that the GLP1 agonists did not improve the cognition or psychosocial function in schizophrenia patients [77]. These conflicts might be due to dosage differences or variations in GLP1 agonists; for example, the differences between Bydureon and Liraglutide [76,77].

## 5. Enteroendocrine Cells in Visceral Pain and Neuropathic Pain

Visceral pain is a severe form of pain originating from the internal organs. However, it is generally difficult to localize. Among heterogenous pathogenic hypotheses, the neurological dysfunction is significantly linked to visceral pain [78,79,80,81,82]. In fact, visceral pain and neuropathic pain are mostly characterized by hypersensitivity to stimulus, potentially due to hypersensitivity of primary sensory afferent neurons and dysregulation of neurotransmission [79,83].

Visceral pain is often correlated with digestive disorders such as IBD or irritable bowel syndrome (IBS) [84,85,86,87]. Due to the complicity of disease pathology, visceral pain sometimes shows resistance to treatment, especially to opioid drugs. In the worst-case scenario, opioid drugs might even worsen the disease symptoms [4,5]. Therefore, an understanding of the novel biological aspects, such as intestinal microbiota and epithelial EECs, in visceral pain would improve the therapeutic treatments. The relationship between the intestinal microbiota and visceral pain modulation has been discussed recently [88,89]. We highlight the potential connections of EECs to visceral pain via unique proteins and peptide hormones, including serotonin, GLP1, PYY, and Guanylate cyclase 2C (GUCY2C) (Figure 4).

Serotonin is predominantly (90%) secreted by ECs in the intestinal epithelium. It could activate the receptors on serotonergic neurons and trigger the enteric nerve system activity for pain [90]. Numerous studies indicate that serotonin signalling is associated with neuron hypersensitivity to pain. An increased number of ECs has been observed in IBS patients, who usually suffer from pain symptoms [91]. Further, the blockage of serotonin signalling by 5-hydroxytryptamine 3 (5-HT3) receptor antagonists reduced pain in IBS patients [92]. Subcutaneous or tissue injection of serotonin induced the hyperalgesia response and interacted with the endocannabinoid system, which further exacerbated pain [93,94,95,96]. The mechanism of serotonin-induced hypersensitivity has been investigated in the past few decades. Serotonin is known to activate 5-HT3 receptors, thus inhibiting the expression of catecholamine-O-methyltransferase (COMT), which contributes to the downregulation of the pain perception and sensitivity [94,97,98,99,100]. Moreover, a recent study provided new evidence of serotonin-mediated visceral hypersensitivity, which worked via 5-hydroxytryptamine 7 (5-HT7) dependent mucosal neurite outgrowth [101]. Therefore, EECs could play important roles in the pathogenesis and severity of visceral pain. Alterations in the characteristics of EECs (especially ECs) might be an effective target for pain treatment. However, the detailed mechanism is still unknown. Future studies are needed to investigate this aspect.

GLP1 is secreted by not only EECs, but also in the CNS system. GLP1 and its receptor have also been suggested to be associated with neuropathic pain and visceral pain in numerous studies, working mainly through the modulation of inflammation. Recent studies showed that the activation of the GLP1/GLP1R axis improved recognition memory impairment, neuroinflammation, and neurological pain via regulating the AMP-activated protein kinase/nuclear factor kappa B (AMPK/NF-κB) pathway [102,103]. Further, the GLP1R agonist decreases pain hypersensitivity through decreasing pro-inflammatory factors and increasing microglia anti-inflammatory factors, such as interleukin 10 (IL-10), cluster of differentiation 206 (CD206), interleukin 4 (IL-4), and arginase 1 (Arg1) [102,104,105,106,107,108]. New research claimed that the gene regulation in response to GLP1R activation is an effective strategy in new treatments for neuropathic pain, by confirming that the GLP1R pathway is involved in pain hypersensitivity mediated by microglia activation [109]. Considering the inter-organ communication though nerve and endocrine systems, regulation of GLP1 and its receptor in the intestine and CNS system could synergistically improve neural pain sensitivity. Similar to neuropathic pain, the GLP1 agonist is also able to decrease visceral pain. In animal models, a GLP1 analogue or GLP1R agonist improved the visceral pain hypersensitivity in rats [110,111]. New evidence in clinical trials has shown that the subcutaneous injection of a GLP1 analogue, ROSE-010, decreased pain hypersensitivity [112,113]. Although these exogenous peptide treatment data strongly support the connection between GLP1 and visceral pain, less research has been performed to understand the effects of endogenous EECs-derived GLP1 in mediating visceral pain. More investigations are needed to identify the potential EECs targets in developing visceral pain treatments.

PYY is mainly expressed in EECs. However, there are only limited data on the relationship between PYY and visceral pain. Neuropeptide Y is in the same family as PYY but is secreted mainly by neurons. Although the neuropeptide Y inhibits the transmission of pain in the spinal cord and brain stem [114], the relationship between PYY and neural pain is still unclear. In IBS patients, PYY cell density was decreased, which has been proposed as a potential biomarker for the disease [115,116]. In a recent study, Hassan et al. used PYY knockout mice to investigate the relationship between pain, PYY, and the Y2 receptor. Their data suggested that the Y2 receptor antagonist and knockout of PYY increased visceral pain [117]. However, future studies are needed to confirm the effect of PYY on visceral pain and to investigate the details of the mechanism.

Finally, the hypothesis of GUCY2C signalling has been linked to visceral pain pathogenesis. A recent study suggested that GUCY2C-enriched intestinal neuropod cells could modulate visceral pain [118]. Further, GUCY2C agonists decreased pain through increasing the cytoplasm cyclic guanosine monophosphate (cGMP) synthesis from guanosine triphosphate (GTP), as well as through releasing the cGMP from the basolateral membrane of the epithelium to the ENS [119,120,121,122,123]. Therefore, GUCY2C agonists have been proposed as a potential treatment for visceral pain (well reviewed previously) [124]. Not surprisingly, GUCY2C is expressed in whole intestinal epithelial cells, including EECs [125,126]. Given the fact that EECs are close to and actively communicate with ENS neurons, one would strongly expect EECs-derived GUCY2C to modulate visceral pain [13,127]. However, there is not yet a clear understanding of EECs’ involvement in GUCY2C-modulated visceral pain. Future research is required to address this.

## 6. Enteroendocrine Cells in Depression

Depression is a common disease that affects up to 350 million people around the world [128]. Although depression is a neurological disease, it has been believed to be related to gastrointestinal disorders. Evidence suggests that constipation is a common comorbidity in depression patients [129]. Further, chronic constipation patients have a 33% of incidence of major depression [129,130]. In depression patients, a high number of ECs has been observed, which indicates a potential relationship between depression, ECs, and serotonin production [91].

In contrast to above hypothesis, serotonin deficiency has long been believed to be one of the potential mechanisms of depression, due to the fact that effective medicines have been serotonin-related [131]. Although a recent systematic review questioned the serotonin deficiency hypothesis due to a lack of sufficient supportive data, it might be due to the heterogenous nature of depression among different studies [132]. In fact, several studies have suggested a relationship between depression and the metabolism of tryptophan, a precursor of serotonin [133,134]. Further, selective serotonin reuptake inhibitor (SSRIs) drugs have been used as treatments for depression patients. Therefore, serotonin is still a potential mechanism and pathway for depression. In addition, recent studies suggested that the modulation of intestinal serotonin metabolism through ECs and oral probiotics improved depressive symptoms in an animal model [14,135,136,137]. The potential mechanism has been proposed to be associated with tryptophan hydroxylase and tryptophan metabolism. For instance, mice fed with probiotics showed increased tryptophan hydroxylase 1 mRNA expression in the colon, and the treatment alleviated depressive behaviour in mice with induced chronic stress [135]. However, more research and evidence are needed to investigate the detailed mechanism of ECs involved in depression.

Recent studies suggest that other EEC products, namely GLP1 and GLP2, are also potentially involved in depression. In stressed mice models, GLP2 played a role in regulating monoamine pathways, which in turn exhibited anti-depressive effects [138,139,140]. Similar to GLP2, GLP1 has also shown anti-depressive effects. Both intraperitoneal administration of a GLP1 analogue (Liraglutide) and or oral delivery of an enhancer (metformin) of endogenous GLP1 secretion have shown an anti-depressive effect in animal models [141,142]. The potential mechanisms of GLP1 in depression have been well reviewed in [143]. Briefly, there are four potential mechanisms of GLP1 involved in depression treatment. First, neuroinflammation is modulated by GLP1. Second, the dysregulation of neurotransmitters is modulated by GLP1. Third, the neurogenesis caused by depression is modulated through GLP1. Finally, GLP1′s regulation of depression induces synaptic dysfunction and memory loss [143]. Given the fact that GLP1 might be produced via a multi-organ system, as well as the dynamically regulated EECs activity in response to the brain–gut–microbiota axis, one should not overlook the potential contribution and significance of EECs-derived GLP1 in modulating depression. Further, since the microbiota metabolites trigger EECs to secrete GLP1 and GLP2, this increases the possibility of EECs serving as mediating regulators between the microbiota and enteric nerve system. However, research is needed to provide evidence to support this hypothesis and investigate the details of the involved mechanism.

## 7. Conclusions

In the present paper we summarized the direct and indirect involvement mechanisms of the EECs in neurological and psychiatric disorders, and discussed the potential treatments. Besides the accumulation of EECs-derived α-synuclein that exacerbates the disease progression in PD, most of the disorders showed significant associations with dysregulation of the neuroendocrine molecules (such as GLP1, GLP2, PYY, serotonin, etc.) produced by EECs and subtype cells. Most of the current treatment strategies focus on administrating exogenous agonists or analogues (GLP1 in schizophrenia, visceral pain hypersensitivity, and depression) and receptor antagonist (serotonin in visceral pain) of these molecules. Alternatively, optimizing the endogenous production of these neuroendocrine molecules could also be considered for developing novel therapeutic strategy. Accumulating evidence connect brain–gut–microbiota axis to the pathogenesis and regulation of neurological and psychiatric disorders. Future investigation should focus on characterizing healthy EECs and reshaping EECs homeostasis in diseases. Intestinal EECs serve as significant source of neuroendocrine molecules. The number and content profiling of EECs depends on intrinsic factors, such as intestinal epithelial stem cells, and the extrinsic microenvironment, such as luminal microbiota. Therefore, the new strategy could be focusing on the differentiation and homeostasis of EECs in the intestinal epithelium, as well as optimizing the EECs functions via regulation of microbiota and nutrition, especially the probiotics and prebiotics. However, the detailed mechanisms are still unclear due to the limitations of techniques and current evidence. The present paper humbly provides a direction for future studies.

## Figures and Tables

**Figure 1 biomedicines-10-02577-f001:**
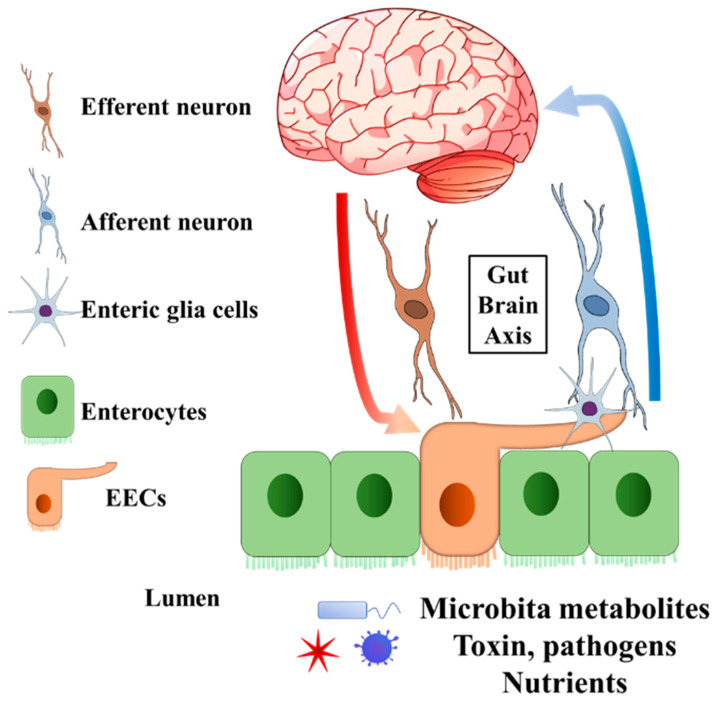
General structure of EECs involved in gut–brain axis. The EECs present receptors on the brush border to sense the microbiota metabolites, toxins, pathogens, and nutrients in the lumen. Enteric glia cells and neurons connect to EECs. The secreted endocrine molecules affect afferent neuron signalling directly and (or) indirectly via EECs enteric glia cells. Efferent neurons bring the signal into the central nervous system. On the other hand, the central nervous system can pass the signal to EECs through efferent neurons. EECs, enteroendocrine cells.

**Figure 2 biomedicines-10-02577-f002:**
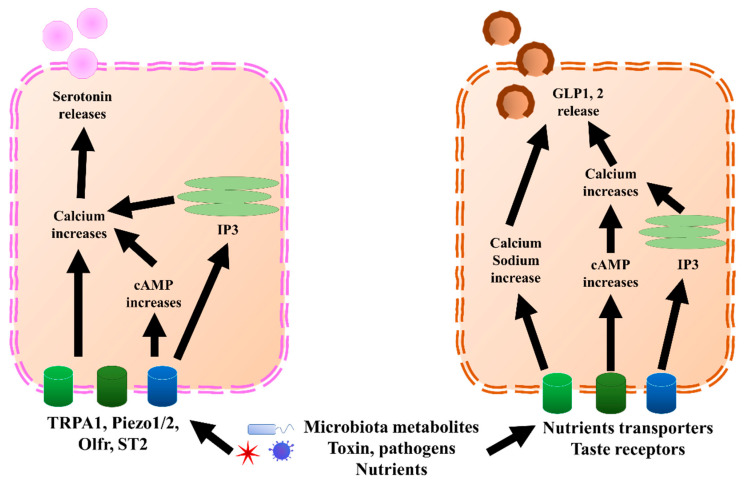
GLP1, 2, and serotonin secretion mechanisms in EECs and ECs, respectively. In ECs, TRPA1, piezol 1/2, Olfr, and ST2 bind to microbiota metabolites, pathogens, or nutrients, which increases calcium levels through directly increasing cAMP or endoplasmic reticulum IP3. Once the calcium level increases, it triggers the release of serotonin. Similar to ECs, EECs sense microbiota metabolites, toxins, pathogens, and nutrients through nutrient transporters and taste receptors. Further, they trigger an increase in calcium and sodium by directly increasing cAMP or endoplasmic reticulum IP3. Once the calcium level increases, it triggers the release of GLP1 and GLP2. cAMP, cyclic adenosine monophosphate; ECs, enterochromaffin cells; EECs, enteroendocrine cells; GLP1, glucagon-like peptide 1; GLP2, glucagon-like peptide 2; IP3, inositol trisphosphate; Olfr, olfactory receptor; TRPA1, transient receptor potential ankyrin 1.

**Figure 3 biomedicines-10-02577-f003:**
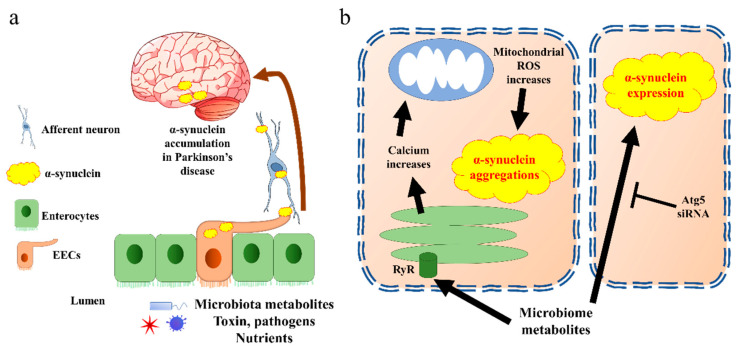
α-synuclein accumulates in Parkinson’s disease through EECs. (**a**) The general pathway by which EECs trigger α-synuclein transfer into brain. Aggregated α-synuclein produced by EECs is transported into brain through afferent neurons and vagal nerve. (**b**) Cell signalling of α-synuclein aggregation present in EECs. The EECs present receptors on the brush border to sense the microbiota metabolites. Triggering of endoplasmic reticulum releases calcium through RyR. The increase in calcium induces reactive oxygen species (ROS) synthesis in mitochondria, which further creates α-synuclein aggregates. Microbiota metabolites also increase α-synuclein expression through Atg5 pathway in EECs. Atg5, autophagy-related 5; EECs, enteroendocrine cells; ROS, reactive oxygen species; RyR, ryanodine receptor.

**Figure 4 biomedicines-10-02577-f004:**
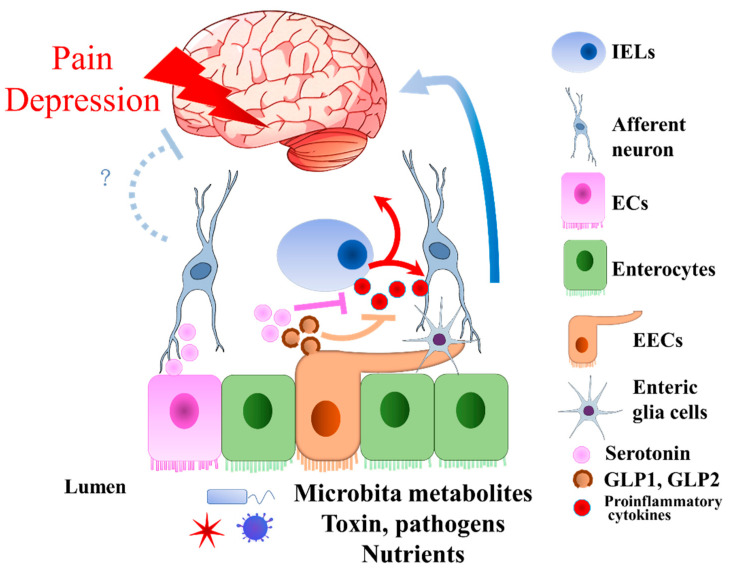
Visceral pain and depression pathologies involved with EECs and ECs. EECs and ECs sense the microbiota metabolites, toxins, pathogens, and nutrients in the lumen to secrete GLP1, 2, and serotonin, respectively. During pathology status, IELs produce proinflammatory cytokines, which enhance the progression of visceral pain and depression. GLP1 and GLP2 secreted by EECs have anti-inflammatory effects. Since the inflammation increases the visceral pain and depression through afferent neurons, the anti-inflammatory effect of GLP1 and GLP2 would decrease the visceral pain and depression. Although it is not clear how serotonin secreted by ECs affects depression through the afferent neurons, it has the effect of anti-inflammation, which might reduce depression. ECs, enterochromaffin cells; EECs, enteroendocrine cells; GLP1, glucagon-like peptide 1; GLP2, glucagon-like peptide 2; IELs, intraepithelial lymphocytes.

## Data Availability

Not applicable.

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
