# Peer review of "Involvement of Intestinal Enteroendocrine Cells in Neurological and Psychiatric Disorders"

_biomedicines, 2022, doi:10.3390/biomedicines10102577_

Round 1

Reviewer 1 Report

The manuscript entitledThe Potential Pathogenic Involvement and 2 Mechanism of Intestinal Enteroendocrine cells in Neurological 3 and Psychological Disorders is well written and organized. But some issues were raised while reading the article. This paper can be reconsidered after careful revision of the following suggestions.

Major Comments:

1.      The abstract needs to be more focused and mention the requirement of this article.

  1. List of abbreviation should be kept in alphabetical order.
  2. The change proposed for the manuscript to have scientific relevance is to make a good methodology following the guidelines of the systematic reviews.
  3. Authors should be used more up-to-date references. Below article along with others could be cited at appropriate place in the revised manuscript.

Upadhayay et al. (2022) Genes, 13: 1324.

Jabir et al. (2021) Annals of Medicine, 53 (1): 2332-2344.

Jabir et al. (2021) Current Pharmaceutical Design, 27 (20): 2425-34.

Suganya, Kanmani, and Byung-Soo Koo. "Gut–brain axis: Role of gut microbiota on neurological disorders and how probiotics/prebiotics beneficially modulate microbial and immune pathways to improve brain functions." International Journal of Molecular Sciences 21.20 (2020): 7551.

Maiuolo, Jessica, et al. "The contribution of gut microbiota–brain axis in the development of brain disorders." Frontiers in neuroscience 15 (2021): 616883.

  1. The paper contains some grammatical mistakes, and some sentence restructuring is required. I will recommend taking the service of language professionals.
  2. The conclusion section should be more specific.

Reviewer 2 Report

Re: Manuscript ID: biomedicines-1868267

The review deals with the involvement of the enteric endocrine cells in some neurologic and psychiatric diseases. The role of enteric nervous system and microbiota in the gut/brain axis was also considered. Anyway, the manuscript is confusing, with several typing and language mistakes, as well as conceptual inexactnesses. The manuscript needs to be completely restyled. In particular, paragraph 2 must be rewritten. Changes are suggested to improve the paper.

Points of criticism

A language restyling of the manuscript is needed.

Title of the article.

Delete “Mini review:” from the title. The title can be shortened: Involvement of Intestinal Enteroendocrine cells in Neurological and Psychiatric Disorders. In the text of the article the authors include enterochromaffin cells, L cells and K cells, as additional and distinguished endocrine cells. To the reviewer it is not clear this approach.

I would replace “psychological” with “psychiatric”.

Why figures 1, 4, 2, 3?

Line 61 of figure 1. Replace “enteroendocrine cells involves” with “EECs involved”.

Lines 61-62 of figure 1. Replace “enteroendocrine cells” with “EECs”.

In figure 1 replace “microbiome” with “microbiota”.

Line 68 of figure 4. What does “ECs” mean? Maybe enterochromaffin cells, as stated in line 86?

Line 83. Elsewhere. Where (references)?

Line 93. Replace “passthrough” with “pass through”

Lines 94-95. What does mean “improved intestinal morphology?

Paragraph 2. This paragraph is entitled “Enteroendocrine cells and functions of peptide hormones” but it also mentions serotonin, which is not a peptide (see also lines 197-198). This paragraph is confusing. The authors mention enteroendocrine cells, enterochromaffin cells and L cells (in the subsequent paragraph – line 119 – K cells are also mentioned). Although this is a mini review, these cells must be clearly distinguished and described. Figure 4 (which should be figure 2) must be included after this paragraph.

Line 117. Replace “an hypothesis” with “a hypothesis”.

Line 161. Replace “increases” with “increase”.

Line 130 and line 160 of the caption of figure 2. ROS abbreviation must be explained.

Lines 172-173. Replace “EECs .” with “EECs.”.

Line 184. Replace “contract” with “contrast”.

Line 199. Serotonins. Why plural?

In figure 3 the role of the microbiota in pain and depression is shown, but it was not described in the text.

Lines 301-303. These sentences are not clear to the reviewer.

Round 2

Reviewer 1 Report

Although, I see some of my comments have been responded by the authors but some of the comments were either ignored or didnt reach to the authors. Hence, I cant recommend this paper to be accepted in its present form.

Author Response

We apologize to Reviewer 1 if we misunderstood any of the comments. However, we have carefully revised manuscript point-to-point according to reviewer's comments. Please kindly help us understand which comments has been ignored.

Reviewer 2 Report

The revised manuscript is now suitable for publication.

Author Response

We thank reviewer for helping improving the manuscript quality.